# The 2-Oxoglutarate Carrier Is S-Nitrosylated in the Spinal Cord of G93A Mutant hSOD1 Mice Resulting in Disruption of Mitochondrial Glutathione Transport

**DOI:** 10.3390/biomedicines11010061

**Published:** 2022-12-27

**Authors:** Daniel A. Linseman, Aimee N. Winter, Heather M. Wilkins

**Affiliations:** 1Department of Biological Sciences, Knoebel Institute for Healthy Aging, University of Denver, Denver, CO 80208, USA; 2Kisbee Therapeutics, Inc., Cambridge, MA 02139, USA; 3Alzheimer’s Disease Research Center, University of Kansas Medical Center, Kansas City, KS 66160, USA

**Keywords:** glutathione, 2-oxoglutarate carrier, Bcl-2, mitochondria, S-nitrosylation, SOD1

## Abstract

Mitochondrial oxidative stress and dysfunction are strongly implicated in the pathogenesis of amyotrophic lateral sclerosis (ALS). Glutathione (GSH) is an endogenous antioxidant that exists as distinct cytosolic and mitochondrial pools. The status of the mitochondrial GSH pool is reliant on transport from the cytosol through the 2-oxoglutarate carrier (OGC), an inner membrane anion carrier. We have previously reported that the outer mitochondrial membrane protein, Bcl-2, directly binds GSH and is a key regulator of OGC-dependent mitochondrial GSH transport. Here, we show that G93A mutant SOD1 (Cu, Zn-superoxide dismutase) reduces the binding of GSH to Bcl-2 and disrupts mitochondrial GSH uptake in vitro. In the G93A mutant hSOD1 mouse model of ALS, mitochondrial GSH is significantly depleted in spinal cord of end-stage mice. Finally, we show that OGC is heavily S-nitrosylated in the spinal cord of end-stage mice and consequently, the GSH uptake capacity of spinal cord mitochondria isolated from these mutant mice is significantly diminished. Collectively, these findings suggest that spinal cord GSH depletion, particularly at the level of the mitochondria, plays a significant role in ALS pathogenesis induced by mutant SOD1. Furthermore, the depletion of mitochondrial GSH in the G93A mutant hSOD1 mouse model may be caused by the S-nitrosylation of OGC and the capacity of mutant SOD1 to disrupt the Bcl-2/GSH interaction, resulting in a disruption of mitochondrial GSH transport.

## 1. Introduction

Amyotrophic lateral sclerosis (ALS) or Lou Gehrig’s disease is a devastating neuromuscular disorder. Clinically, ALS is characterized by progressive skeletal muscle weakness and atrophy, leading eventually to respiratory failure and death typically within 2–5 years of diagnosis [1]. Pathologically, ALS is characterized by a progressive degeneration of motor neurons in the motor cortex, brainstem and spinal cord and corresponding retraction of motor axons away from the neuromuscular junctions [2,3]. Although the precise mechanisms underlying neurodegeneration and disease progression are unknown, evidence indicates that mitochondrial dysfunction and mitochondrial oxidative stress (MOS) are significant contributing factors in the death of motor neurons [4]. Mitochondria are primarily responsible for ATP production and are metabolically dynamic organelles that generate significant amounts of reactive oxygen species (ROS) as a by-product of electron transport. Dysfunctional mitochondria and elevated levels of MOS have been implicated in the pathogenesis of several neurodegenerative disorders including ALS [5]. The accumulation of oxidative stress in mitochondria leads to further damage of mitochondrial components resulting in a vicious feed forward cycle of dysfunction.

Mutations in the antioxidant enzyme, copper/zinc superoxide dismutase (SOD1), have been implicated in mitochondrial dysfunction and are a major cause of familial ALS [6,7,8]. The G93A mutant human Cu, Zn-superoxide dismutase (G93A mutant hSOD1) mouse model of familial ALS develops an aggressive form of motor neuron disease reminiscent of human ALS [9,10]. These mice have been shown to develop several clinical and histopathological features of ALS including muscle atrophy, hind limb weakness and paralysis, loss of spinal cord motor neurons, and markers of neuroinflammation [9,10]. Several studies in the G93A mutant hSOD1 mouse have provided evidence for MOS and mitochondrial dysfunction. For instance, antioxidants and free radical scavengers targeted to the mitochondria mitigate the degeneration of motor neurons in these mice [11,12]. Glutathione (GSH) is an abundant endogenous tripeptide antioxidant with distinct cytosolic and mitochondrial pools. Depletion of GSH has been observed in whole spinal cord tissue and spinal cord mitochondria of end-stage G93A mutant hSOD1 mice; however, the mechanism underlying mitochondrial GSH depletion has not been investigated [13,14,15].

Enzymes required for the synthesis of GSH are not found in the mitochondria, therefore GSH is exclusively synthesized in the cytosol and transported into the mitochondria [16]. GSH is transported into the mitochondria via a facilitated transport process involving mitochondrial inner membrane anion transporters such as dicarboxylate (DIC), 2-oxoglutarate (OGC) and tricarboxylate carriers [17,18,19]. Transport of GSH into the mitochondria from the cytosol is critical to maintaining mitochondrial function. Overexpression of OGC in a rat renal proximal tubular cell line (NRK-52E) enhanced mitochondrial GSH transport and showed significant protective effects against chemically induced apoptosis [20]. Previously, we have shown in primary cerebellar granule neurons that mitochondrial GSH transport is critical for cell survival and inhibition of DIC sensitizes these neurons to oxidative and nitrosative stress [21]. We have also shown that stable overexpression of the OGC transporter in NSC34 motor neuron-like cells was sufficient to increase mitochondrial GSH levels and provide protection from oxidative and nitrosative stress [22]. Furthermore, studies have shown the anti-apoptotic protein Bcl-2 to regulate cellular GSH levels [23]. Overexpression of Bcl-2 conveys an increase in cellular GSH levels, while conversely Bcl-2 knockout mice show a significant reduction in GSH levels [24,25]. Previously, we have shown that GSH transport through OGC is facilitated by a synergistic interaction with Bcl-2 resulting in enhanced transport of GSH into the mitochondria. A direct interaction between Bcl-2 and GSH facilitates GSH transport through OGC and provides protection from oxidative stress [22,26,27].

Here, we show that the Bcl-2/GSH interaction and GSH uptake in isolated rat brain mitochondria are both enhanced by wild type (WT) SOD1 but perturbed in the presence of G93A mutant SOD1 recombinant proteins in vitro. Disruption of the Bcl-2/GSH interaction results in a reduced capacity to transport GSH into the mitochondria. Using isolated mitochondria from the spinal cords of end-stage G93A mutant hSOD1 mice, we show a reduction in the mitochondrial GSH levels corresponding to a disruption of mitochondrial GSH uptake. Furthermore, we show that the OGC is heavily S-nitrosylated in mitochondria isolated from the spinal cords of end-stage G93A mutant hSOD1 mice, further disrupting transport of GSH into the mitochondria.

## 2. Materials and Methods

### 2.1. Bcl-2/GSH Binding Assay

Recombinant Bcl-2 (50 ng) was incubated with agarose beads (Ag), GSH agarose beads (GSHAg), or GSHAg in the presence of either 25 ng WT SOD1 or 25 ng G93A SOD1 with mixing by rotation for 1 h at 4 °C. Bead complexes were washed and proteins were resolved by SDS-PAGE and immunoblotted for Bcl-2, as previously described [27]. Human recombinant Bcl-2 protein (His-tagged) was purchased from Oncogene Research Products (San Diego, CA, USA). WT SOD1 and G93A SOD1 human recombinant proteins (His-tagged) were obtained from Acro Biosystems (Newark, DE, USA).

### 2.2. Isolation of Rat Brain Mitochondria and GSH Transport Assays

Mitochondria were isolated from Sprague Dawley rat brains as previously described [27]. Isolated rat brain mitochondria were incubated with 2 mM GSH in the presence of vehicle (Con), 25 ng WT SOD1, or 25 ng G93A SOD1 for 4 h. Incubations were performed in GSH transport buffer (5 mM HEPES, pH 7.2, 220 mM mannitol, 70 mM sucrose, 0.1 mM EDTA, 0.1% BSA (fatty acid-free), 5 mM succinate, and 1 mM potassium phosphate) at room temperature. Samples were washed, and total glutathione was measured using a DTNB assay kit from Oxford Biomedical, following the manufacturer’s protocol. All GSH measurements were normalized to protein concentration.

### 2.3. G93A Mutant hSOD1 Mouse Model of ALS

All animal procedures were performed according to protocols approved by the University of Denver Institutional Animal Care and Use Committee (IACUC). FVB-Tg (SOD1-G93A) mice with the toxic gain of function SOD1 Gly position 93 to Ala mutation (G93A) were obtained from The Jackson Laboratory (Bar Harbor, ME, USA). The colony was maintained and bred in the animal facility at the University of Denver. Animal genotyping was done by the third-party company, Transnetyx (Cordova, TN, USA). These mice are a widely used model of familial ALS and develop an aggressive form of motor neuron disease with a well characterized disease progression. We have previously shown that mice housed in our breeding facility develop disease onset predictably at approximately 90 days of age and progress to end-stage of disease at approximately 120 days of age [15]. These mice have also been shown to develop several clinical and histopathological features of ALS such as loss of motor neurons, neuroinflammation, hind limb weakness, and skeletal muscle atrophy [9]. Littermates of both sexes (equal number of males and females) were included in each group, with an age- and sex-matched non-transgenic (NonTG) littermate mouse as the control. End-stage was defined as the point at which a mouse placed on its side failed to right itself to a sternal position within 15 s. When it was determined that a mouse had reached end-stage, the animal was immediately euthanized by inhaled isoflurane overdose and secondary decapitation.

### 2.4. Isolation of Mitochondria from Lumbar Spinal Cord and Cerebral Cortex

Mitochondria were isolated as previously described [15]. Briefly, lumbar spinal cord or cerebral cortex was homogenized using a dounce homogenizer in 2 mL of mitochondrial isolation buffer (0.64 M sucrose, 2 mM EDTA, 20 mM Tris-HCl, pH 7.4). Samples were placed in 24% Percoll gradient and centrifuged at 4 °C for 5 min, 16,000 rpm to make a 12% Percoll gradient. Two milliliters of supernatant were then added to a Percoll gradient containing a bottom layer of 2 mL 40% Percoll and a top layer of 2 mL 19% Percoll. Samples were spun at 4 °C for 10 min, 16,000 rpm. The mitochondrial layer was removed (2 mL of the layer between the 19% and 40% gradients, respectively), washed with mitochondrial buffer at a dilution of 4:1 and spun at 4 °C for 10 min at 11,820 rpm. The supernatant was discarded, and the pellets were washed a second time with 4.5 mL of mitochondrial isolation buffer and 0.5 mL of mitochondrial isolation buffer containing 5 mg/mL BSA, then spun at 4 °C for 10 min at 7500 rpm. Mitochondrial pellets were then re-suspended in 100 μL of a mitochondrial buffer containing 130 mM KCl, 4 mM Tris-HCl, 5 mM sodium pyruvate, 5 mM sodium succinate and 1 mM EGTA, pH 7.4.

### 2.5. GSH Measurement by HPLC-EC

For HPLC-electrochemical detection (EC) analysis, mitochondria isolated from lumbar spinal cord or cerebral cortex were added to 10% PCA prior to vortexing 3 times for 15 s each time. The supernatant was removed and filtered. Mitochondrial GSH values were normalized to protein concentrations, measured by BCA protein assay kit (Thermo Scientific, Rockford, IL, USA). Samples were analyzed in triplicate, and an additional sample was spiked with reduced GSH and glutathione disulfide (GSSG) to verify correct peak identification. HPLC-EC conditions were as previously described and total GSH represents reduced GSH + GSSG [15].

### 2.6. Detection of S-Nitrosylation of OGC

Whole lumbar spinal cord from NonTg and end-stage G93A mutant hSOD1 mice was lysed and immunoprecipitated with an antibody to OGC. The immunoprecipitated OGC was modified using the Iodo-TMT S-nitrosylation kit from Pierce ThermoFisher as per the manufacturer’s recommendations. Modified OGC protein was resolved via SDS-PAGE and immunoblotted for TMT. Separate samples of whole spinal cord lysates were immunoblotted for OGC and beta tubulin (as a loading control). Western blots were performed as previously described [21].

### 2.7. Statistical Analysis

The results shown represent the means ± S.E.M. for the number (*n*) of independent experiments performed. Statistical differences were obtained either by one-way analysis of variance followed by a post hoc Tukey’s test or by an unpaired Student’s *t* test. A *p* value of <0.05 was considered statistically significant. In general, immunoblots shown are representative of the results obtained in at least three independent experiments.

## 3. Results

### 3.1. The Bcl-2/GSH Interaction and Mitochondrial GSH Uptake Are Both Enhanced by WT SOD1 but Perturbed in the Presence of G93A Mutant SOD1 Recombinant Proteins

We first examined the effects of recombinant WT or G93A mutant SOD1 proteins on the Bcl-2/GSH interaction and mitochondrial GSH uptake in vitro. We observed that recombinant G93A mutant SOD1 diminishes the interaction between Bcl-2 and GSH (Figure 1A), while recombinant WT SOD1 slightly enhances the Bcl-2/GSH interaction (Figure 1B). Quantification shows a significant reduction in the Bcl-2/GSH interaction in the presence of mutant G93A SOD1 when compared to the WT SOD1 recombinant protein (Figure 1C).

Next, freshly isolated rat brain mitochondria were incubated with 2 mM GSH for 4 h under control conditions or in the presence of either recombinant WT SOD1 protein (hSOD1WT) or recombinant G93A mutant SOD1 protein (hSOD1G93A). While WT SOD1 significantly enhanced mitochondrial GSH uptake into isolated mitochondria, G93A mutant SOD1 trended towards a reduction in mitochondrial GSH loading compared to control conditions (Figure 2). In comparison to the WT SOD1 protein, mitochondria incubated with the G93A mutant SOD1 protein displayed a significantly lower capacity to transport GSH.

### 3.2. Mitochondrial GSH in Lumbar Spinal Cord Is Depleted in End-Stage G93A Mutant hSOD1 Mice

We examined GSH levels within the G93A mutant hSOD1 mouse model of ALS. Mitochondrial total glutathione (GSH + GSSG) from lumbar spinal cord and cerebral cortex were measured using HPLC coupled to electrochemical detection. Spinal cord mitochondria from end-stage G93A mutant hSOD1 mice had significantly depleted GSH compared to mitochondria from NonTg littermate controls (Figure 3). In contrast, mitochondria from transgenic mouse cortex did not show a significant difference in GSH content compared to NonTg controls.

### 3.3. Mitochondria Isolated from Lumbar Spinal Cord of End-Stage G93A Mutant hSOD1 Mice Are Deficient in Their Capacity to Take Up GSH In Vitro

Mitochondria isolated from lumbar spinal cord or cerebral cortex were incubated in vitro with 2 mM GSH for 4 h and the amount of GSH taken up over time was measured. Mitochondria isolated from lumbar spinal cord of end-stage G93A mutant hSOD1 mice took up significantly less GSH compared to mitochondria isolated from age-matched, NonTg control mice (Figure 4). Mitochondria isolated from cortex of the same end-stage G93A mutant hSOD1 mice showed no significant difference in the amount of GSH loaded compared to NonTg controls.

### 3.4. Mitochondria Isolated from Mouse Lumbar Spinal Cord Depend on OGC for GSH Transport

We next determined if lumbar spinal cord mitochondria were dependent on OGC for GSH transport. Mitochondria isolated from NonTg mouse lumbar spinal cord were incubated with either the DIC inhibitor, butylmalonate (BM), or the OGC inhibitor, phenylsuccinate (PS). No significant difference was observed between mitochondrial GSH uptake under control conditions or in the presence of BM, although a trend towards decreased uptake was apparent (Figure 5). In contrast, addition of PS significantly reduced GSH uptake by nearly 70% compared to controls (Figure 5). These results indicate that spinal cord mitochondria are largely dependent on OGC for GSH transport.

### 3.5. OGC Is S-Nitrosylated in Lumbar Spinal Cord from End-Stage G93A Mutant hSOD1 Mice

Reduced capacity of lumbar spinal cord mitochondria isolated from end-stage G93A mutant hSOD1 mice to transport GSH could be a consequence of several effects. First, these transgenic mice may express reduced protein levels of key mitochondrial GSH transporters in spinal cord. However, we observed no significant difference in OGC protein levels in lumbar spinal cord of end-stage G93A mutant hSOD1 mice when compared to age-matched NonTg control mice (Figure 6A). Modification of the OGC in mitochondria may account for the reduced capacity to transport GSH. We measured the S-nitrosylation of OGC within lumbar spinal cord of end-stage G93A mutant hSOD1 mice. End-stage G93A mutant hSOD1 mice displayed a significant increase in the amount of TMT-labeled OGC protein in lumbar spinal cord compared to NonTg mice indicating S-nitrosylation of cysteine residues (Figure 6B). Furthermore, addition of GSH to the immunoprecipitated OGC protein prevented the subsequent TMT labeling, consistent with a blockade of S-nitrosylation (Figure 6B).

## 4. Discussion

Mitochondria lack the enzymes necessary to synthesize GSH and therefore must rely on transporters, such as OGC, to import GSH from the cytosol [16]. Transport of GSH from the cytosol into mitochondria is essential for cell survival. Based on our previously published work [21,22,26,27] and the findings presented here, we conclude that OGC-dependent mitochondrial GSH transport is compromised in mitochondria of G93A mutant hSOD1 mice. The Bcl-2/GSH interaction and mitochondrial uptake of GSH is disrupted in the presence of G93A mutant SOD1 recombinant protein in vitro. Furthermore, mitochondria isolated from the spinal cord of end-stage G93A mutant hSOD1 mice show reduced GSH levels and a diminished capacity to take up GSH in vitro, effects that are associated with a marked S-nitrosylation of the OGC transporter.

Previous studies have shown the ability of G93A mutant SOD1 to interact with Bcl-2 at mitochondria [28,29]. Indeed, Pedrini et al. showed that G93A mutant SOD1, but not WT SOD1, induces a conformational change in Bcl-2 that exposes its pro-apoptotic BH3 domain [29]. We have shown that GSH-binding by Bcl-2 facilitates an association with OGC to enhance OGC-dependent mitochondrial GSH transport [22,26,27]. With this in mind, we aimed to determine the effect of WT SOD1 and mutant SOD1 on the Bcl-2/GSH interaction. We observed that recombinant G93A mutant SOD1 diminishes the interaction between Bcl-2 and GSH, while recombinant WT SOD1 slightly enhances the Bcl-2/GSH interaction. Furthermore, incubation of rat brain mitochondria with WT SOD1 enhances GSH transport into the mitochondria when compared to controls; however, this effect is lost in mitochondria incubated with G93A mutant SOD1. These data suggest a novel function of WT SOD1 to enhance the Bcl-2/GSH interaction and facilitate mitochondrial GSH transport. In contrast, G93A mutant SOD1 appears to interfere with GSH-binding by Bcl-2 and consequently, decreases mitochondrial GSH transport. The precise mechanism underlying these differential effects of WT and G93A mutant SOD1 is currently unknown. Mutant G93A SOD1 retains its dismutase activity but is known to generate more hydroxyl radicals than WT SOD1 via a gain-of-function mechanism [30]. It is possible that hydroxyl radical production in the presence of G93A mutant SOD1 could reduce the capacity of isolated mitochondria to transport GSH. However, we favor an alternative explanation for this effect in that G93A mutant SOD1, but not WT SOD1, induces a conformational change in Bcl-2 as described previously by Pedrini et al. [29], and that this conformational change reduces the capacity of Bcl-2 to interact with GSH and facilitate OGC-dependent mitochondrial GSH transport.

We next examined the mitochondrial pool of GSH within the spinal cord of G93A mutant hSOD1 mice at end-stage compared to NonTg littermate controls. Our results are consistent with previous reports [13,14,15] and show that total mitochondrial GSH is significantly depleted in the lumbar spinal cord of end-stage G93A mutant hSOD1 mice. This reduction in the mitochondrial pool of GSH may be a direct result of deficient mitochondrial GSH transport as mitochondria isolated from spinal cord of end-stage G93A mutant hSOD1 mice displayed a deficit in this function in vitro. We further found that spinal cord mitochondria required OGC for GSH transport and that the OGC transporter is heavily S-nitrosylated in spinal cord from end-stage G93A mutant hSOD1 mice.

We hypothesize that the deficit in mitochondrial GSH transport observed in the spinal cord of end-stage G93A mutant hSOD1 mice may arise from multiple effects. First, mutant SOD1 disrupts the Bcl-2/GSH interaction resulting in reduced GSH uptake through the OGC transporter. Second, it has been previously reported that Bcl-2 expression is reduced, and/or Bcl-2 function is compromised, within the spinal cord of ALS patients and G93A mutant hSOD1 mice [31,32]. Given our data establishing an important regulatory role for Bcl-2 in OGC-dependent mitochondrial GSH transport [22,26], loss of Bcl-2 function could play a role in diminishing mitochondrial GSH transport in the mutant SOD1 mouse model of ALS. Third, oxidative modifications to OGC could directly impair mitochondrial GSH transport. In particular, OGC has three cysteine residues which play critical roles in its structure and function making this transporter a prime target for inactivation by S-nitrosylation [17].

Our findings implicate the deficiency in spinal cord mitochondrial GSH transport observed in the G93A mutant hSOD1 mouse model as a potential contributing factor in ALS pathogenesis by either directly promoting motor neuron cell death or facilitating the neuroinflammatory phenotype of glial cells. In future studies, it would be interesting to evaluate mitochondrial GSH levels and mitochondrial GSH transport in skeletal muscle which is equally involved in ALS and in which oxidative stress also plays a detrimental role [33,34]. Furthermore, we predict that the deficiency in spinal cord mitochondrial GSH transport limits the therapeutic actions of known mitochondrial “protective” compounds and that rescue of OGC-dependent mitochondrial GSH transport could act synergistically with mitochondrial protective agents to significantly delay disease progression and prolong survival of G93A mutant hSOD1 mice. Identification of OGC-dependent mitochondrial GSH transport as a key deficiency in diverse models of ALS would reveal a novel site for therapeutic intervention. Moreover, strategies aimed at preserving OGC function and sustaining the mitochondrial GSH pool may represent new effective approaches to protect motor neurons in ALS.

## Figures and Tables

**Figure 1 biomedicines-11-00061-f001:**
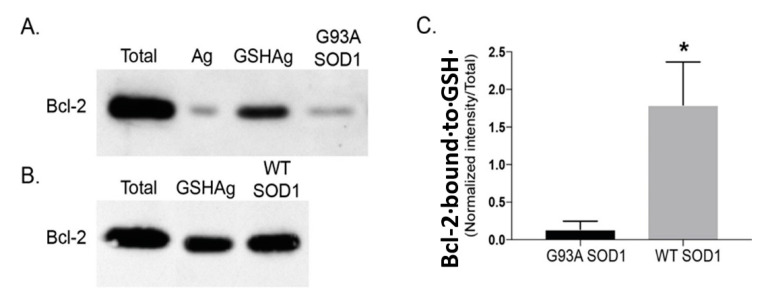
G93A mutant SOD1 decreases Bcl-2 binding to GSH. (**A**) Recombinant Bcl-2 (50 ng) was incubated with agarose beads (Ag), GSH agarose beads (GSHAg), or GSHAg in the presence of 25 ng G93A SOD1, and proteins were resolved by SDS-PAGE and immunoblotted for Bcl-2. (**B**) Recombinant Bcl-2 was incubated with GSH agarose beads (GSHAg) alone or in the presence of 25 ng WT SOD1, and proteins were resolved by SDS-PAGE and immunoblotted for Bcl-2. (**C**) Quantification of Bcl-2 bound to GSH showing a reduced binding in the presence of mutant G93A SOD1 recombinant protein. Mean ± SEM; * *p* < 0.05, unpaired Student’s *t*-test (*n* = 3 independent experiments).

**Figure 2 biomedicines-11-00061-f002:**
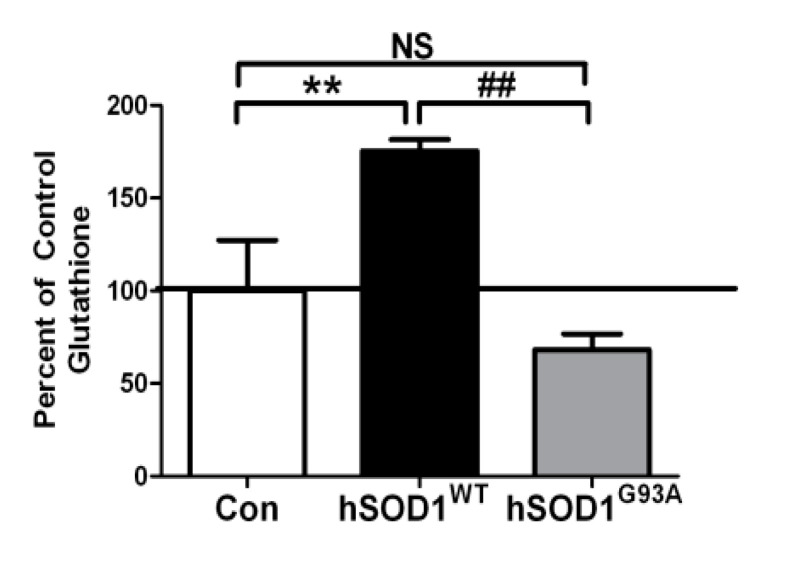
Mitochondrial GSH loading is enhanced with WT SOD1 but diminished with G93A mutant SOD1 in isolated rat brain mitochondria. Isolated rat brain mitochondria were incubated with 2 mM GSH in the presence of vehicle (Con), 25 ng hSOD1WT, or 25 ng hSOD1G93A for 4 h. Samples were washed and total glutathione was measured using a DTNB assay. Mean ± SEM; NS = not significant, ** *p* < 0.01 versus Con, ## *p* < 0.01 versus hSOD1WT via a One-way ANOVA with a post hoc Tukey’s test (*n* = 3 independent experiments).

**Figure 3 biomedicines-11-00061-f003:**
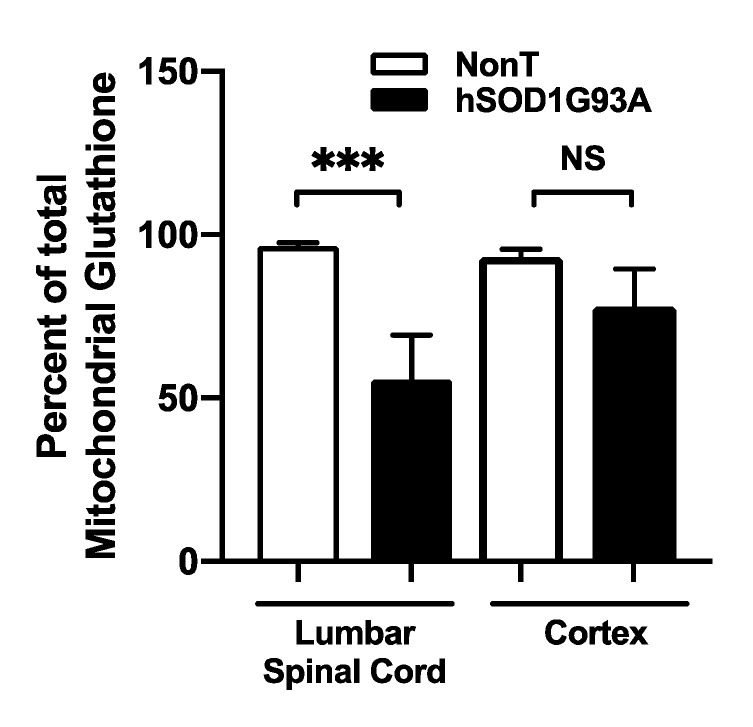
GSH is diminished in mitochondria isolated from G93A mutant hSOD1 spinal cord. Mitochondria were isolated from NonTg and G93A mutant hSOD1 mice at end-stage (~120 days-old) from lumbar spinal cord and cerebral cortex, and total GSH was measured using HPLC-EC. All measurements were normalized to protein content and represented as a percent of NonTg. Mean ± SEM; NS = not significant, *** *p* < 0.001, as determined by unpaired Student’s *t*-test, *n* = 4.

**Figure 4 biomedicines-11-00061-f004:**
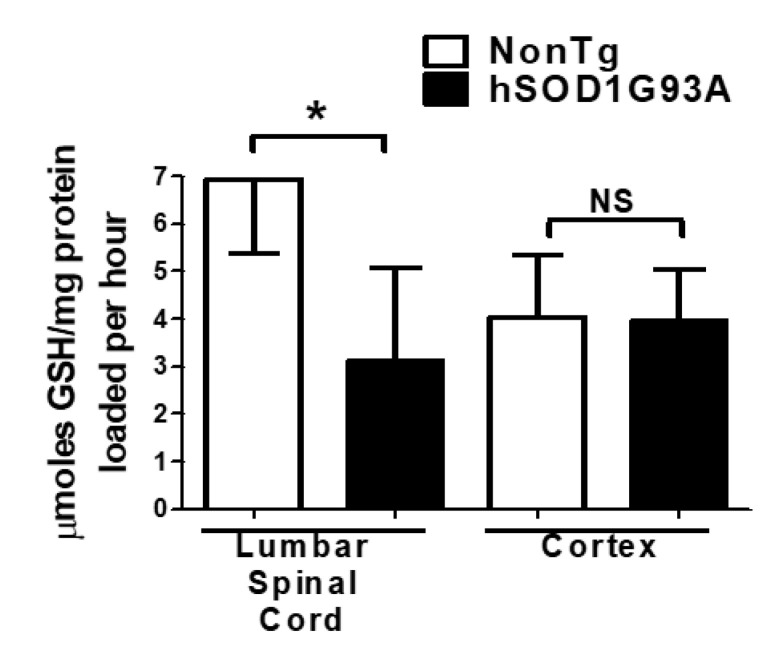
GSH loading is diminished in mitochondria isolated from G93A mutant SOD1 spinal cord. Isolated lumbar spinal cord or cerebral cortex mitochondria from G93A mutant hSOD1 and littermate NonTg control mice were incubated with 2 mM GSH for 4 h. Samples were washed and GSH was measured using a DTNB assay. Mean ± SEM; NS = not significant, * *p* < 0.05 versus NonTg control, as determined by unpaired Student’s *t*-test, *n* = 4.

**Figure 5 biomedicines-11-00061-f005:**
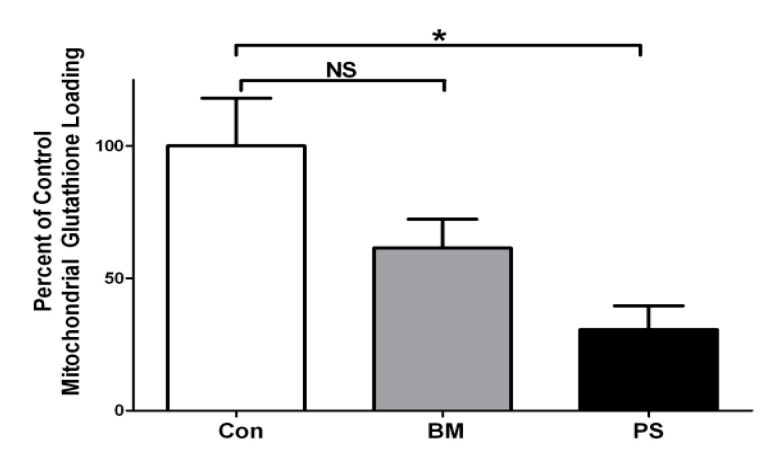
Mitochondria isolated from mouse lumbar spinal cord are dependent on OGC for GSH transport. Isolated mouse (NonTg) lumbar spinal cord mitochondria were incubated with 2 mM GSH in the presence of either vehicle (Con), 5 mM butylmalonate (BM), or 5 mM phenylsuccinate (PS) for 4 h. Samples were washed and total GSH was measured using a DTNB assay. Mean ± SEM; * *p* < 0.05 versus Con, NS = not significant, via a one-way ANOVA with post hoc Tukey’s test; *n* = 4.

**Figure 6 biomedicines-11-00061-f006:**
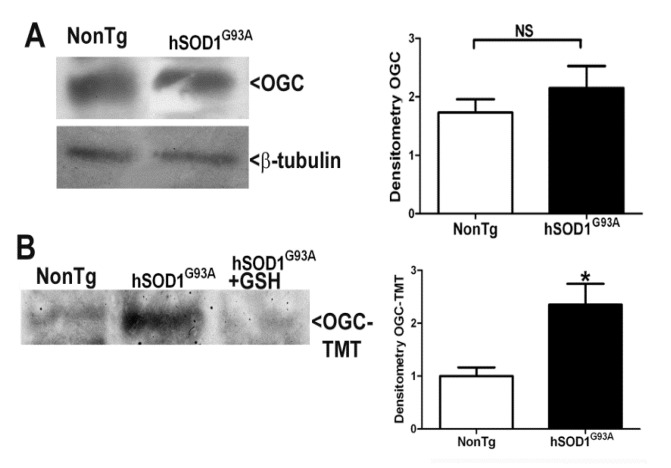
OGC is S-nitrosylated in lumbar spinal cord from end-stage G93A mutant hSOD1 mice. (**A**) Lumbar spinal cord tissue from non-transgenic (NonTg; FVB/NJ) and end-stage G93A mutant hSOD1 mice (hSOD1G93A) was lysed and equal protein amounts were resolved by SDS-PAGE and immunoblotted against OGC and β-tubulin. Densitometry was completed to measure OGC band intensity normalized to β-tubulin. NS = not significant via an unpaired Student’s *t*-test, *n* = 4. (**B**) Whole lumbar spinal cord from NonTg and end-stage hSOD1G93A mice was lysed and immuno-precipitated (IP’d) for OGC. IP’d OGC was modified using the Iodo-TMT S-nitrosylation kit from Pierce ThermoFisher as per the manufacturer’s recommendations. IP’d and modified protein was resolved via SDS-PAGE and immunoblotted for TMT. GSH was added to one reaction as a negative control. Densitometric analysis of TMT-labeled OGC was performed. * *p* < 0.05 versus NonTg via an unpaired student’s *t*-test, *n* = 4.

## Data Availability

Data are available by request from the corresponding author.

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
