# Peer review of "The 2-Oxoglutarate Carrier Is S-Nitrosylated in the Spinal Cord of G93A Mutant hSOD1 Mice Resulting in Disruption of Mitochondrial Glutathione Transport"

_biomedicines, 2022, doi:10.3390/biomedicines11010061_

Round 1
Reviewer 1 Report
This study shows that the Bcl-2/GSH interaction and GSH uptake in isolated mitochondria are both enhanced by wild type (WT) SOD1 but perturbed in the presence of G93A mutant SOD1 recombinant proteins in vitro. Disruption of the Bcl-2/GSH interaction results in a reduced capacity to transport GSH into the mitochondria. Using isolated mitochondria from the spinal cords of end-stage G93A mutant hSOD1 mice, the mitochondrial GSH levels were reduced. The authors also show that the OGC is heavily S-nitrosylated in mitochondria isolated from the spinal cords of end-stage G93A mutant hSOD1 mice, further disrupting transport of GSH into the mitochondria.
The experiments were carried out properly and the results are presented in a concise manner.
The authors should pay attention to the points below.
G93A mutation is a pseudo one that leaves the enzyme activity as superoxide dismutase.
Therefore, one question arises; what is the mechanism(s) underlying the difference between G93A mutant and WT. This could be one of the fundamental questions for those numerous studies using G93A mutant. The mutant G93A SOD1 is known to increase hydroxyl radical production due to its gain-of-functon. Does hydroxyl radical play a role in the observed results in this study? Or could it be due to changes in structural requirements for interacting OGC or Bcl-2? Inclusion of these insights may make this manuscript more significant.
Reviewer 2 Report
The manuscript of Linseman et al. is original and well conducted. The authors describe the mitochondrial involvement in ALS and in particular that mitochondrial GHS is reduced in spinal cord of end-stage SOD1 G93A mice. The obtained results suggest a new contributing factor in ALS pathogenesis.
However, there are some points to clarify.
-Transgenic male and female were used in which percentage?
-Detail on the mouse model should be added in the Method section
-I think it should be better explained what means end-stage mice? Each animal show symptoms? They are walking? Did you observe a loss of righting reflex at this stage?
-Why in Fig.2 are used rat brain mitochondria and not the mouse as the same species?
-It could be interesting to evaluate the same effect (i.e. GHS measure) in mitochondria of skeletal muscle which is equally involved in ALS and in which ROS are detrimental (doi.org/10.1038/s41598-019-39676-3 14; doi:10.1152/ajpcell.00372.2013)?
